# Bioequivalence, Drugs with Narrow Therapeutic Index and the Phenomenon of Biocreep: A Critical Analysis of the System for Generic Substitution

**DOI:** 10.3390/healthcare10081392

**Published:** 2022-07-26

**Authors:** Lucia Gozzo, Filippo Caraci, Filippo Drago

**Affiliations:** 1Clinical Pharmacology Unit, Regional Pharmacovigilance Centre, University Hospital of Catania, 95125 Catania, Italy; f.drago@unict.it; 2Department of Biomedical and Biotechnological Sciences, University of Catania, 95125 Catania, Italy; 3Centre for Research and Consultancy in HTA and Drug Regulatory Affairs (CERD), University of Catania, 95125 Catania, Italy; fcaraci@unict.it; 4Department of Drug and Health Sciences, University of Catania, 95125 Catania, Italy; 5Unit of Translational Neuropharmacology, Oasi Research Institute-IRCCS, 94018 Troina, Italy

**Keywords:** generics, bioequivalence, substitution, biocreep, narrow therapeutic index drugs

## Abstract

The prescription of generic drugs represents one of the main cost-containment strategies of health systems, aimed at reducing pharmaceutical expenditure. In this context, most regulatory authorities encourage or obligate dispensing generic drugs because they are far less expensive than their brand-name alternatives. However, drug substitution can be critical in particular situations, such as the use of drugs with a narrow therapeutic index (NTI). Moreover, generics cannot automatically be considered bioequivalent with each other due to the biocreep phenomenon. In Italy, the regulatory authority has established the Transparency Lists which include the medications that will be automatically substituted for brand-name drugs, except in exceptional cases. This is a useful tool to guide prescribers and guarantee pharmaceutical sustainability, but it does not consider the biocreep phenomenon.

## 1. Introduction

Once the patent of brand-name medications expires, generic equivalents can enter the market. Thanks to the reduced cost, the use of generics represents a major cost-containment measure, aimed at reducing expenditure for public health systems [1,2,3,4].

Generics are pharmaceutically equivalent to the reference products, being medicines with the same qualitative and quantitative composition in the active substance(s), the same pharmaceutical form and route of administration. However, they may differ in terms of excipients or additional constituents which potentially affect drug stability, absorption, and toxicity [1].

To obtain approval, manufacturers which develop generics need to demonstrate the product bioequivalence (BE) with the brand-name equivalent by appropriate bioavailability studies [1,5,6].

A product can be considered bioequivalent to the brand-name drug if after administration of the same dose, it exhibits a similar degree and rate of absorption [7]. These features guarantee the comparability of two drug preparations in terms of in vivo efficacy and safety and make them interchangeable and replaceable [4,7,8,9]. Since efficacy studies are not necessary, some clinicians and patients have expressed concerns about the therapeutic equivalence of generic drugs.

Nevertheless, evidence from multiple studies demonstrated the therapeutic equivalence between generics and brand-name medications across multiple disease areas, even from real-world evidence. The study of Desai RJ et al. based on data from healthcare databases of more than 3.5 million patients showed analogous clinical outcomes with the use of generics and the brand-name products for chronic diseases including diabetes, hypertension, osteoporosis, and psychiatric disorders [10].

Regarding the use of cardiovascular drugs, both generic and brand-name products demonstrated similar effectiveness in terms of risk of hospitalization for cerebrovascular and coronary events [11,12,13] and comparable drug-related adverse events [14]. Corrao et al. demonstrated that the risk of events is not changed by substituting antihypertensive drugs with their generics or vice versa [11]. Similarly, patients treated with statins who started or switched to generics during follow-up did not experience a different risk of discontinuation nor of cardiovascular (CV) outcomes compared to those starting on brand-name medications, including no significant difference in the risk of hospitalization for major events [13,15].

This evidence invalidates the widespread theory of the clinical superiority of brand-name drugs compared to generic drugs in cardiovascular diseases [12].

However, this does not mean that generics can automatically be considered bioequivalent each other.

## 2. Requirements for Approval of Generics and Criteria for Drug Substitution

While the efficacy and safety profile of the reference products must be supported by data from preclinical and clinical studies, the approval for generics requires substantially reduced supporting evidence [1,5,16].

The definition of bioequivalence varies somewhat among regulatory authorities. The US Food and Drug Administration (FDA) defines BE as the ‘absence of a significant difference between two or more products in the rate and extent of absorption at the site of drug action when administered at the same molar dose under similar conditions’ [17]. The European Medicines Agency (EMA) defines generic as ‘a medicinal product which satisfies the criteria of having the same qualitative and quantitative composition in terms of active substances, of having the same pharmaceutical form, and of being bioequivalent, unless it is apparent in the light of scientific knowledge that it differs from the original product with regard to safety and efficacy’ [7]. The evaluation of BE is based on the maximum plasma concentration (Cmax) and the area under the curve (AUC) of plasma drug concentration over time after administration, two pharmacokinetic (PK) parameters which describe the rate and extent of drug absorption [18].

According to BE criteria, the variability in Cmax and AUC can range from −20% to +25% between the generic and the reference product [1]. These criteria are recognized by several regulatory authorities worldwide (Table 1) and if met, manufacturers are not required to submit additional safety and efficacy data to support therapeutic equivalence [19,20,21,22,23].

Given the general regulatory position that bioequivalence translates into therapeutic equivalence, generics may be interchangeable or replaceable with the reference product [24].

To contain drug costs, most regulatory authorities encourage or obligate the substitution of drug products with the most affordable versions, generally included on a specific list, according to a positive formulary approach [25,26].

In different national contexts, pharmacists may automatically substitute generic and reference products on the formularies and/or medical doctors are obligated to fill the prescription only with the active substance of the product [27]. Automatic generic substitution includes the possibility to freely switch among different generic versions of the same medication.

For example, generic substitution is compulsory in a lot of European countries, such as Sweden and Germany, and several measures have been implemented to enhance prescribing generics to obtain significant savings and more efficient use of healthcare resources [28,29,30,31]. Generic prescribing represents a measure of high quality prescribing in the UK, ref. [31] which resulted in savings of GBP millions in four main therapeutic areas (statins, proton pump inhibitors, drugs that affect the renin-angiotensin system, and clopidogrel). This goal has been achieved thanks to incentives put in place to encourage prescribing generics versus originators but also through a high rate (more than 80%) of prescribing by chemical name, attributable to physicians’ training and education [28,30,31,32,33].

Nevertheless, regulatory authorities allow some exceptions concerning an agreed list of medicines excluded from substitution because of their pharmacological properties or patients’ clinical condition [34].

Moreover, it is noteworthy that different generic versions of a single reference product may show such PK variability to represent a critical issue in case of a switch from one generic to another due to the so called ‘biocreep’ phenomenon (Figure 1).

For example, two generics are both bioequivalent with the reference product with a plasma drug level 25% greater and 20% lower than the reference, respectively. Switching from the first to the second product without dose adjustments could result in a significant reduction in plasma drug levels and a potential loss of efficacy; in contrast, switching from the second to the first product could increase plasma drug levels with a possible reduction in terms of tolerability. This is of particular concern in the treatment of specific disorders, such as some neurological and neuropsychiatric diseases and in general in case of drugs with a narrow therapeutic index (NTI) [35].

## 3. Drugs with Narrow Therapeutic Index: Bioequivalence versus Therapeutic Equivalence

The goal of any pharmacological treatment is to achieve the desired therapeutic effect without producing adverse effects in any patients.

The therapeutic index (TI) is the relationship between the therapeutic dose and the toxic dose of a drug. It is the ratio (TD50/ED50) between the dose required to produce a toxic effect in 50% of the population (TD50) and the dose required to produce a therapeutic effect in 50% of the population (ED50). A therapeutic ratio of two means that a two times higher dose will cause toxicity in half of the patients than the dose needed to produce a therapeutic effect in the same proportion of patients. [36].

Therefore, this index quantifies the relative safety of a drug and the larger the TI, the safer the drug is. In the case of NTI, it can be difficult to use a product in clinical practice without a strict monitoring of plasma concentration to prevent unacceptable toxicities or treatment failure. For drugs with an NTI, also known as ‘critical dose drugs’, small variations in plasma concentrations can result in an insufficient therapeutic response or onset of adverse effects, and therefore their use should be individualized, particularly in patients of older age, with comorbidities or receiving multiple medications (Table 2) [37,38,39,40].

In case of an NTI, the doses to use are chosen to provide the highest benefit with the lowest risk, considering both the nature of the potential negative effects and the potential therapeutic effects.

With regard to the development of generics, many agencies choose stricter criteria for bioequivalence definition for NTI drugs to face the potential ineffectiveness and adverse effects related to PK variability (Table 1). The acceptance interval for BE is tightened to 90–111% from the standard 80–125% levels. This general rule is not enough to guarantee treatment safety and effectiveness in some cases, above all for drugs with a within-subject variability moderate to high (frequently exposed to bioavailability differences larger than 10% with the same product) [41,42]. In the case of drugs with high interindividual variability, the risk of treatment complications cannot be solved; only applying stricter BE criteria and drug substitution can be dangerous for patients [43,44]. Indeed, even a small variability of plasma concentration for the so-called critical dose drugs (as defined by the Canadian Drug Regulatory Agency) can lead to serious therapeutic failures and/or serious adverse drug reactions [45].

There is not a universally accepted list of NTI drugs, and not all regulatory agencies have developed a set of criteria to define NTI products. According to EMA guidelines, it is not possible to define a list of specific criteria to classify NTI drugs, and it must be decided case by case based on clinical considerations [7]. Even the FDA does not provide a comprehensive list, but NTI medications requiring stricter bioequivalence can be identified from product-specific guidance. For example, carbamazepine, everolimus, phenytoin, warfarin, digoxin, and valproic acid are considered NTI drugs that are subject to stricter bioequivalence requirements [46].

Because of the narrow range between safe and lethal doses, some experts expressed concerns about the therapeutic equivalence of the generics of these products and proposed to not apply substitution of generics for NTI drugs [43,47,48].

Concerns about the therapeutic equivalence are well represented by the experience with antiepileptics. Bioequivalence studies demonstrate that generic and branded antiepileptics are interchangeable, but conflicting literature results have raised doubts in both patients and clinicians about therapeutic equivalence and indiscriminate drug substitution [49].

Several clinical reports showed a recurrence of seizures in well-controlled patients or the increase of seizure frequency following a switch from a reference drug to a generic version related to a substantial change in serum drug levels. [48,50]. Lack of therapeutic equivalence can be explained at least in part by differences in the manufacturing process (due to, e.g., modification of the site of manufacture) which may result into bioavailability changes [51]. Nevertheless, quality defects in single batches are possible despite mandatory monitoring of the quality of all medicines to ensure that only high-quality products are available, both for originators and generics [52]. In general, when patients are well-controlled with the lowest dose of an antiepileptic, the switch among drugs is not recommended [53].

However, differences between old and new antiepileptics have been described, and medicines belonging to specific categories should be considered individually [54,55,56,57,58].

In particular, specific guidance has been issued by the UK Medicines and Healthcare products Regulatory Agency (MHRA) to help prescribers decide whether it is necessary to maintain a specific product or whether switching may be acceptable [59,60]. The guidance identifies three categories of risk:Category 1, including phenytoin, carbamazepine, phenobarbital and primidone; switching between versions of these antiepileptics is not recommended;Category 2, including sodium valproate, lamotrigine, perampanel, retigabine, rufinamide, clobazam, clonazepam, oxcarbazepine, eslicarbazepine, topiramate and zonisamide; the decision about prescription is based on clinical status (such as seizure frequency and treatment history) and patient/caregiver point of view;Category 3, including levetiracetam, lacosamide, tiagabine, gabapentin, pregabalin, ethosuximide, brivaracetam and vigabatrin; in this case, patients can be freely switched among different versions of antiepileptics.

In addition to the three risk categories, additional factors should be considered, such as patient anxiety, risk of confusion, or medication errors.

Pregabalin deserves special attention. It belongs to the category suitable for substitution and is an example of all those drugs with multiple therapeutic indications and different patent protection [34]. Indeed, pregabalin is approved for epilepsy and generalized anxiety disorder, but a new indication with patent protection has been approved, namely neuropathic pain. This situation created the basis for legal actions by the company to protect the brand-name product in the event of non-compliance with the patent conditions, both for the brand name and for future generics. 

Furthermore, it has been proposed to exclude certain patient groups from substitution of generics, including patients with a higher risk of unfavorable outcomes, such as potentially serious complications, treated with polymedication, with complex dosage regimens, or in the case of a high risk of interactions [61].

For example, substitution of immunosuppressive drugs can be critical in transplant patients due to the risk of rejection [62]. However, normal clinical practice requires the monitoring of plasma concentrations during initial treatment with immunosuppressants and after switching, minimizing the risk of failure.

Another therapeutic class with a high risk of under/over-treatment consequent to drug substitution is represented by the central analgesics. Indeed, opioids are characterized by a NTI, a wide interindividual variability in terms of response which differs among opioid-tolerant patients and opioid-naïve patients and can lead to potentially life-threatening toxicity [63].

Opioids are usually used to treat acute pain related to trauma, diagnostic and surgical procedures, and other acute medical problems; moreover, they are also used for the treatment of moderate and severe chronic pain, in cancer and noncancer patients with no adequate response to nonopioid drugs [64,65,66]. Chronic pain is a complex disease entity with a major impact on healthcare costs that results from a maladaptive functional and structural transformation process, sustained by mechanisms of peripheral and central sensitization involving altered neuronal activity [67]. In this scenario, opioids are an essential pharmacological tool for the prevention and management of chronic pain and central sensitization. Two basic criteria should be met when selecting the analgesic strategy: therapeutic appropriateness and timing. In particular, the selected drug must target the main mechanisms responsible for the particular type of chronic pain, while tackling central sensitization and controlling the nociceptive and neuropathic components [68].

When considering therapeutic appropriateness, a balance between adequate pain relief and an acceptable safety profile must be guaranteed. A rapid change in the drug plasma levels may jeopardize this balance, especially when it is urgent and essential in clinical practice to prevent the transition from acute to chronic pain [69,70]. The pharmacodynamic profile of different opioids should be considered for individualized, tailored, mechanism-based therapy with the more potent the opioid, the greater may be the clinical consequences of changes in absolute doses as well as the impact in terms of safety [71].

Opioid therapy produces a spectrum of adverse effects including sedation, hypotension, and respiratory depression linked to the analgesic effect and have abuse potential that has been recently reconsidered [66,72,73]. Primary causes of unwanted clinical effects observed with opioid prescriptions are improper use by the patient or improper prescription by clinicians [66,74].

It has been demonstrated that bioavailability issues with opioid generics may cause insufficient pain relief on the one hand and an increased risk of adverse effects on the other hand [75,76]. The worst consequence of too low dosing because of brand changes would be inadequate analgesia, whereas the switch to a generic with higher bioavailability would result in the onset of serious adverse events, including respiratory depression in extreme situations, which represents the most dangerous adverse effect and can be fatal [5,76]. The clinical impact of an inappropriate switching to a generic might be also increased when considering the rotation from an opioid to another opioid with an equivalent analgesic activity, a pharmacological strategy currently adopted in clinical practice for the management of chronic pain [77].

Brand changes and switching to a generic equivalent opioid with an inappropriate tapering of the new “equivalent” opioid might result in inadequate analgesia, especially in individuals with chronic pain who use long-term opioid therapy (LTOT) [78]. Furthermore, the chronic use of opioids can lead to the development of tolerance and dependence, with consequent further narrowing of the drug’s therapeutic index and increased risk of side effects [69].

Finally, we should consider that almost all opioid analgesics are metabolized by the drug-metabolizing enzymes of the cytochrome P450 (CYP450) system [63]. Opioid analgesics, as all the drugs with an NTI, show an increased risk of clinically relevant drug–drug interactions (DDIs), both pharmacodynamic and pharmacokinetic, that can make therapy management even more complex [79,80,81]. Given the common use of opioids in the context of a polypharmacotherapy, many patients are concurrently prescribed drugs that could precipitate a DDI such as CYP2D6/CYP3A4 inhibitors or inducers that can result in either an excess of opioid levels effects (above the minimal toxic concentration) or a reduction in opioid effect (plasma levels below the minimal effective concentration), respectively [82]. Concomitant treatment with specific CYP2D6/CYP3A4 inhibitors or inducers can therefore cause significant changes in the plasma concentrations of opioid analgesics [82]. When considering the relatively restricted margin of safety of opioids, the consequences of such kinetic modifications might become clinically relevant in particular after brand changes and switching to a generic equivalent opioid as observed with transdermal fentanyl [76]. This can explain why the German Federal Institute for Drugs and Medical Devices restricts the exchangeability of pain relief patches to preparations having equal delivery rates, application intervals, and total amount of drug [83].

## 4. The Italian Regulation for Prescription of Generic Drugs: Strengths and Limits

In Italy, physicians and pharmacists must comply with the requirements set out in specific guidelines concerning prescription rules [84].

In the case of ongoing treatments of medicinal products not reimbursed by the national health system (NHS) or in the lack of generics of the product, physicians may always prescribe a specific medicine. On the other hand, physicians should fill in the prescription by indicating the name of the active substance if the patient is being treated for the first time for a chronic disease or is being treated for a new episode of a non-chronic disease if generics of the product are on the market.

Moreover, doctors can prescribe a specific product if it is considered not replaceable for a specific patient [85], but the non-replaceability clause must be necessarily and properly justified. The prescription will be non-compliant with law if the motivation is not specified or in the case of inappropriate motivation.

According to the same Decree [84], the pharmacist should act as follows:If the prescription only indicates the active substance, the pharmacist informs the patient and will deliver the medicine with the lowest price fully covered by the NHS [86]; if the patient expressly requests a product with a higher price, the price differential between the medicine requested and the one with the lowest price will be paid out-of-pocket;If the prescription indicates a specific medication without the non-replaceability clause, the pharmacist can provide the prescribed medicine only if no cheaper equivalent is available; otherwise, the pharmacist must supply the medicine with the lowest price, unless the patient requests and pays for the brand-name product;Finally, if the prescription indicates the non-replaceability condition, the pharmacist should ask the patient to pay the difference between any higher price of the prescribed medicine.

In addition, it should be remembered that biosimilars are considered interchangeable (by doctors) but not replaceable (by pharmacists) according to the Italian legislation [87].

AIFA publishes the monthly Transparency Lists which include the medicinal products having equal composition (same active substances, method of administration, number of posological units, and equal unit doses) and the relative reference prices which establish the costs of generic and brand-name medications. [86,88,89,90,91,92].

Medications included in the Lists and belonging to the same group are automatically replaceable, excluding exceptional cases for technical-scientific reasons, such as the therapeutic index or other specific features of the product [93]. Substitutability depends on the bioequivalence of the generic with the corresponding brand-name medication. However, bioequivalence is only a surrogate for the therapeutic equivalence of two medications, and drug substitution is recognized as critical in particular situations, such for NTI drugs [7,24,94,95,96,97].

Indeed, the Italian regulation still provides for exceptional cases in which some molecules or categories are not included in the List and therefore are not automatically inter-changeable. It is well known a low therapeutic index can cause serious adverse reactions if the replacement of the originator with a generic (or vice versa) or switching among generics may result in variations of the bioavailability of the active substance. In other cases, substitution may result in variations in the lack of bioavailability leading to therapeutic failure. For example, in 2007 AIFA excluded antiepileptics, central analgesics, and anticoagulants from the Transparency List based on their therapeutic index to ensure therapeutic continuity for fragile sub-populations. In case the prescribed medication is not substitutable, the patient will not pay any difference in price.

In Italy, the first prescription of levetiracetam and topiramate [98] should be filled with an equivalent drug, at an advantageous cost for the NHS. In contrast, for epileptic patients whose symptoms are completely controlled by pharmacological therapy, it is recommended not to substitute the drug used, regardless of whether it is the reference or the generic. For patients not completely controlled by drug therapy but who have had significant improvements in terms of frequency or type of seizures, substitutions are also not recommended. 

The Italian Transparency Lists represent a useful tool supporting healthcare professionals for prescription and dispensing of drugs, with an eye toward controlling medication costs to make pharmaceutical expenditure sustainable. However, a major issue is that the current framework does not consider the critical phenomenon of biocreep. Instead, it makes decisions on substitution of bioequivalents based on economic principles. This represents a major risk for patients in terms of medication effectiveness and therapeutic safety.

Furthermore, greater efforts in the direction of ‘transparency’ are needed, defining clear shared rules for inclusion/non-inclusion of drugs in the Lists, [93].

It would be of high importance to establish a clear set of pharmacological features and clinical conditions which would lead to the exclusion from the Lists.

Finally, it would be desirable to go beyond the prescribing obligations (and punishments) to specific training programs for healthcare professionals with the aim of obtaining a high-level education, both scientific and regulatory. Even patient education and scientific/regulatory dissemination with appropriate language for non-experts should be improved to build an appropriate culture surrounding pharmacology and health policy issues.

## Figures and Tables

**Figure 1 healthcare-10-01392-f001:**
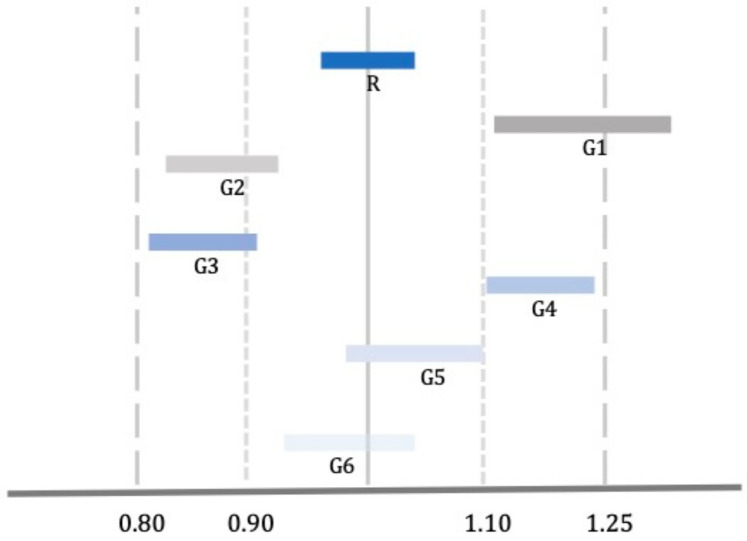
Comparison among reference (R) and generic drugs (G). G1 is not bioequivalent to R according to the standard criteria (−20/+25), and G2 is not bioequivalent to R according to the stricter criteria for NTI drugs. G3 and G4 meet the standard criteria of bioequivalence; however, they are not bioequivalent to each other. G5 and G6 meet the criteria of bioequivalence for NTI drugs.

**Table 1 healthcare-10-01392-t001:** Criteria for regulatory approval of generics.

Agency	Bioequivalence Criteria	Bioequivalence Criteria for Narrow Therapeutic Index Drugs
European Medicines Agency (EMA)	80.00–125.00%	90.00–111.11%
U.S. Food and Drug Administration (FDA)	80.00–125.00%	90.00–111.11%
Japanese Institute of Health Sciences (National Institute of Health Sciences, NIHS)	80.00–125.00%	90.00–111.11%
Health Protection and Food Branch (HPFB) of Canada	80.00–125.00%	90.00–112.00%

**Table 2 healthcare-10-01392-t002:** Examples of drugs with narrow therapeutic index.

Category	Drugs
Anticoagulants	Vitamin K antagonists, Heparin
Antiepileptic drugs	Valproic acid, Phenobarbital, Phenytoin, Carbamazepine
Aminoglycosides	Streptomycin, Kanamycin, Netilmicin, Tobramycin, Neomycin
Immunosuppressant	Cyclosporine, Sirolimus, Mycophenolic acid
Glycoside	Digoxin, Digitoxin
Mood-stabilizing agent	Lithium carbonate
Central analgesics	Opioids

## Data Availability

Not applicable.

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
