# Peer review of "Bioequivalence, Drugs with Narrow Therapeutic Index and the Phenomenon of Biocreep: A Critical Analysis of the System for Generic Substitution"

_healthcare, 2022, doi:10.3390/healthcare10081392_

Round 1
Reviewer 1 Report
The manuscript presented by Gozzo et al., review the bioequivalence among narrow therapeutic index drugs. In my point of view the manuscript is interesting and can be accepted for publication.
I would recommend only a more critical discussion, as claimed in the title, regarding the BE criteria for NTI drugs. One or two paragraphs would be ok.
The authors may explore the issue that the BE range 90-111% for NTI drugs is not enough to ensure clinical equivalence.
Also, a figure similar to Figure 1 could be included for different NTI drugs, since the criteria is different (90.00–111.00% ) with examples cited in the references of the review
Author Response
Dear Reviewer
thank you for your comments and suggestions. We updated the manuscript accordingly.
- I would recommend only a more critical discussion, as claimed in the title, regarding the BE criteria for NTI drugs. One or two paragraphs would be ok. The authors may explore the issue that the BE range 90-111% for NTI drugs is not enough to ensure clinical equivalence.
We updated the manuscript accordingly.
- Also, a figure similar to Figure 1 could be included for different NTI drugs, since the criteria is different (90.00–111.00% ) with examples cited in the references of the review
The figure includes the example of NTI drugs and we described in detail different classes of drugs with NTI in the text, in particular antiepileptics and opioids.
Reviewer 2 Report
The theme of the manuscript entitled “Bioequivalence, drugs with narrow therapeutic index and the phenomenon of biocreep: a critical analysis of the system for generic substitution” is good. However, the manuscript needs major revision as suggested below.
1. The manuscript lacks a clear objective of writing this article, data collection part (methodology), discussion, and conclusion part. The manuscript must be written clearly with separate headings.
2. A lot of typo, grammar, and formatting errors have been identified.
3. Examples of drugs showing biocreep phenomenon is not well described.
4. The details of at least one case study related to the drug having narrow therapeutic index must be provided (Table 2).
5. The full form of some abbreviations is not clear, for example, Table 1 (Does FDA means USFDA? Does NIHS actually represents Japanese Institute of Health Sciences?
6. The authors states that “Once patent of branded products expires, generics can enter the market (line 28). Please note that the entry of a generic drug into the market also depends on the marketing exclusivity granted by the regulatory agencies even in the absence of any patent, for example, new chemical entity exclusivity or orphan drug exclusivity. Accordingly, the authors must modify the written sentence.
Author Response
Dear Reviewer
thank you for your comments and suggestions. we updated the manuscript accordingly.
- The manuscript lacks a clear objective of writing this article, data collection part (methodology), discussion, and conclusion part. The manuscript must be written clearly with separate headings.
This manuscript is not an original article, therefore it has been designed as a Perspective/review, according to the Instructions for authors.
- A lot of typo, grammar, and formatting errors have been identified.
We updated the manuscript accordingly.
- Examples of drugs showing biocreep phenomenon is not well described.
The biocreep phenomenon is a generic concept which therefore does not concern a specific category of medicinal products or a particular active substance It is due to the lack of direct comparison of pharmacokinetic parameters among generics of the same drug. Such PK variability represents a critical issue in case of switch from one generic to another without dose adjustments, with high risk of loss of efficacy or reduction in terms of tolerability.
- The details of at least one case study related to the drug having narrow therapeutic index must be provided (Table 2).
We described in detail different classes of drugs with NTI in the text, in particular antiepileptics and opioids.
- The full form of some abbreviations is not clear, for example, Table 1 (Does FDA means USFDA? Does NIHS actually represents Japanese Institute of Health Sciences?
We updated the manuscript accordingly.
- The authors states that “Once patent of branded products expires, generics can enter the market (line 28). Please note that the entry of a generic drug into the market also depends on the marketing exclusivity granted by the regulatory agencies even in the absence of any patent, for example, new chemical entity exclusivity or orphan drug exclusivity. Accordingly, the authors must modify the written sentence.
We updated the manuscript accordingly.
Round 2
Reviewer 2 Report
None